# Case Report of an Injectional Anthrax in France, 2012

**DOI:** 10.3390/microorganisms8070985

**Published:** 2020-06-30

**Authors:** Jean-Marc Thouret, Olivier Rogeaux, Emmanuel Beaudouin, Marion Levast, Vincent Ramisse, Fabrice V. Biot, Eric Valade, François Thibault, Olivier Gorgé, Jean-Nicolas Tournier

**Affiliations:** 1Centre Hospitalier Centre Hospitalier Métropole Savoie, rue Lucien Bizet, 73000 Chambéry, France; Jean-Marc.Thouret@ch-metropole-savoie.fr (J.-M.T.); olivier.rogeaux@ch-metropole-savoie.fr (O.R.); Emmanuel.beaudouin@ch-metropole-savoie.fr (E.B.); Marion.Levast@ch-metropole-savoie.fr (M.L.); 2DGA Maîtrise NRBC, 5 rue Lavoisier, 91710 Vert le Petit, France; vincent.ramisse@intradef.gouv.fr; 3CNR-LE Charbon (National Reference Laboratory for Anthrax), Institut de Recherche Biomédicale des Armées, 1 Place Général Valérie André, 91220 Brétigny sur Orge, France; Fabrice.Biot@intradef.gouv.fr (F.V.B.); eric1.valade@intradef.gouv.fr (E.V.); Francois.thibault@def.gouv.fr (F.T.); Olivier.gorge@intradef.gouv.fr (O.G.); 4Département Microbiologie et Maladies Infectieuses, Institut de Recherche Biomédicale des Armées, 1 Place Général Valérie André, 91220 Brétigny sur Orge, France; 5Direction Centrale du Service de Santé des Armées, 60 Boulevard du Général Martial Valin, 75015 Paris, France; 6Ecole du Val-de-Grâce, 1 Place Alphonse Laveran, 75 005 Paris, France

**Keywords:** injectional anthrax, drug-user, outbreak, phylogeny

## Abstract

(1) Background: *Bacillus anthracis* is a spore-forming, Gram-positive bacterium causing anthrax, a zoonosis affecting mainly livestock. When occasionally infecting humans, *B. anthracis* provokes three different clinical forms: cutaneous, digestive and inhalational anthrax. More recently, an injectional anthrax form has been described in intravenous drug users. (2) Case presentation: We report here the clinical and microbiological features, as well as the strain phylogenetic analysis, of the only injectional anthrax case observed in France so far. A 27-year-old patient presented a massive dermohypodermatitis with an extensive edema of the right arm, and the development of drug-resistant shocks. After three weeks in an intensive care unit, the patient recovered, but the microbiological identification of *B. anthracis* was achieved after a long delay. (3) Conclusions: Anthrax diagnostic may be difficult clinically and microbiologically. The phylogenetic analysis of the *Bacillus anthracis* strain PF1 confirmed its relatedness to the injectional anthrax European outbreak group-II.

## 1. Introduction

*Bacillus anthracis* is a spore-forming, Gram-positive bacterium causing anthrax, a zoonosis affecting mainly livestock [1]. When infecting humans, it provokes classically three different clinical forms: cutaneous, digestive and inhalational anthrax. A fourth injectional form in intravenous drug users was described in 2000 [2]. So far, only two large outbreaks of injectional anthrax have been described, affecting mainly Northern European countries in two separate waves spanning 2009–2010 and 2012–2013, causing 70 cases and 26 fatalities [3]. We report here the clinical and microbiological features as well as the phylogenetic analysis of the strain responsible for the only injectional anthrax case observed in France so far.

## 2. Case Presentation 

Three days after two heroin injections at the right elbow groove (one intravenous and another probably subcutaneous), a male patient consulted in June 2012 at the outpatient emergency department of a clinic. At the initial clinical examination, the patient presented a mildly painful edema of the right forearm. He was apyretic, and the inflammation biological markers were normal. He was initially discharged but was recalled the next day as the blood cultures grew for a Gram-positive bacterium, considered initially as a *Bacillus spp*. The patient was chilling with an extensive edema of the right upper arm, without sign of necrotizing fasciitis. An anterior fasciotomy of the entire right upper limb was performed to prevent any compartment syndrome. Biological markers clearly suggested a bacterial infection (major leukocytosis at 29 G/L including 82% granulocytes, C reactive protein at 61.2 mg/L, procalcitonin at 0.15 µg/L). An antibiotic therapy associating clindamycin and piperacillin-tazobactam was initiated. The patient was subsequently transferred to an intensive care unit (ICU) at the Centre Hospitalier de Chambéry, France after the onset of a septic shock associated with a major dermo-hypodermatitis without necrotizing fasciitis. The patient received intensive resuscitation for three weeks for a shock associated with coagulation-induced abnormalities needing iterative transfusions. He required external dialysis due to an acute renal failure. Antibiotics were switched to the association of tazocillin and clindamycin. The edematous syndrome was clinically major with very significant hydro-electrolytic losses by aponevrotomy incisions (up to 10 L/day at the climax) and with pleural and pericardial effusions necessitating a drain. The right arm dressings were regularly made at the operating room. The use of local negative pressure therapy was possible after two weeks. The antibiotic therapy was finally switched to amoxicillin-clavulanic acid. The patient was discharged from ICU after 3 weeks at the time of the *B. anthracis* identification. The local condition improved slowly after the fourth week, allowing the antibiotics to be discontinued. A skin graft was performed at the fifth week on the forearm and the arm. At nine months, the only sequela was a radial paralysis due to a nerve posterior compression.

## 3. Results

### 3.1. Laboratory Identification

The initial blood cultures identified were a *Bacillus cereus,* despite the use of two different techniques (Vitek 2 BCL and API 50 CH (Biomérieux, Marcy l’Etoile, France). An Institut National de Veille Sanitaire (INVS) alert about a European injectional anthrax outbreak led to seeking a more precise identification. A 16S rRNA PCR sequencing confirmed the presence of a *Bacillus cereus* group member (including *B. anthracis*) at the time the patient was discharged from the ICU (26 days after infection). The final identification of strain PF1 was confirmed at the national reference laboratory for anthrax (Figure 1).

At the examination, Gram-positive “jointed bamboo-rod” bacilli were identified. The bacilli were non-hemolytic, immobile and capsulated. *B. anthracis* was rapidly confirmed by real-time PCR performed with in-house probes targeting *CapC* and *PagA* by using a GeneXpert (Cepheid, Sunnyvale, CA, USA) and a LightCycler 2.0 (Roche, Basel, Switzerland). The MALDI-TOF analysis performed on a MALDI Biotyper (Bruker Daltonics, Billerica, MA, USA) with an extended bioterrorism agent database was consistent with molecular identification, with a high score of 2.301 for *B. anthracis*. In vitro antibiotic testing by Etest^®^ (Biomérieux, Marcy l’Etoile, France) confirmed this isolate susceptibility to penicillin, doxycycline, ciprofloxacin and levofloxacin.

### 3.2. Phylogeny Analysis

DNA library was prepared with Nextera XT kit (Illumina), sequenced on an MiSeq instrument (Illumina, San Diego, CA, USA). Library quality control was done with FastQC (Available online: http://www.bioinformatics.babraham.ac.uk/projects/fastqc/). Sequencing produced 9,550,862-paired reads with a Guanosine/Cytosine (GC) percentage of 36%. Raw data were imported in Bionumerics software v7.6 (Applied Maths) and for performing all subsequent analyses for single-nucleotide polymorphism (SNP) detection and phylogenetic. Ames reference sequence was downloaded from GenBank while the other strain sequences were downloaded from SRA [4]. Paired-end reads were aligned against the Ames ancestor genome, and SNPs were called through the “Strict SNP filtering (closed SNP set)” filter, with a minimal 10x coverage and minimum interval of 12 base pairs between two SNP. Phylogenetic analysis was performed by minimum spanning tree (MST). European injectional anthrax cases clustered in two distinct genotypic groups: Group I for the 2009–2010 patients, and Group II for the 2012–2013 outbreak [4]. Whole-genome SNP analysis identified our isolate as belonging to the Group II cluster (Figure 2). 

The MST supported the hypothesis of a micro-evolution within Group II from the oldest known injectional anthrax strain isolated in Norway in 2000, leading after three mutation events to the six-strain cluster (two Danish, one German and three UK strains), from which emerged five derivatives, all except one differentiated by only one SNP. Only the UR-1 strain harbored two SNPs compared to the cluster. Our isolate exhibited only one derived SNP (position 2,546,129) compared to the cluster of six identical genomes that represented the ancestral genome. It shared this SNP with UR-1 but missed the second derived SNP found in UR-1 (position 2,438,242) [5]. Thus, our strain filled a gap between the ancestral genome and the German UR-1 strain, both sharing a common evolution path. Raw reads were deposited in the NCBI SRA with accession number SAMN13173321.

## 4. Discussion and Conclusions

Our report completes the description of this 2012–2013 European outbreak encompassing four different countries with the description and the publication of the strain responsible for the last injectional anthrax case [3].

For this patient, the diagnosis was initially delayed, highlighting the challenging determination of the exact species in the *Bacillus cereus* group by routine analysis. In our case, the epidemiological alert led to the exact identification of the strain, although it did not change the therapeutic based on an adapted antibiotherapy combined with non-specific supportive therapy. Eventually, this patient recovered from this severe dermohypodermatitis and the development of a drug-resistant shock. The cause of the shock may have been multiple, including sepsis, hypovolemia and coagulopathy, as well as the massive production of *B. anthracis* toxins. As suggested by the extensive size of the edema and the positivity of the blood culture, the toxin levels may have been very high. We have shown in preclinical models that elevated blood toxin levels start even very early on during the course of an infection [6,7]. Unfortunately, a retrospective analysis of the toxin levels in the serum and the hydro-electrolytic losses was not possible for this patient. The administration of adjunct anti-toxin therapy was evaluated in the Scotland cohort with an ambiguous success [8,9], and this would have probably not changed the prognosis.

The origin of the strain has been already discussed in detail [4]. So far, efforts to detect anthrax spore contamination in heroin batches by culture or PCR have failed [10]. In this case, no testing of the drug bought in Switzerland was possible. The phylogenetic analysis ascribed the strain in the Group II of the second wave of the European outbreak, confirming the genetic link and the same origin of the outbreak.

Anthrax diagnostics may be difficult clinically and microbiologically, especially for the atypical injectional form of anthrax.

## 5. Ethic Statement

Patients provided permission to publish clinical data anonymously.

## Figures and Tables

**Figure 1 microorganisms-08-00985-f001:**
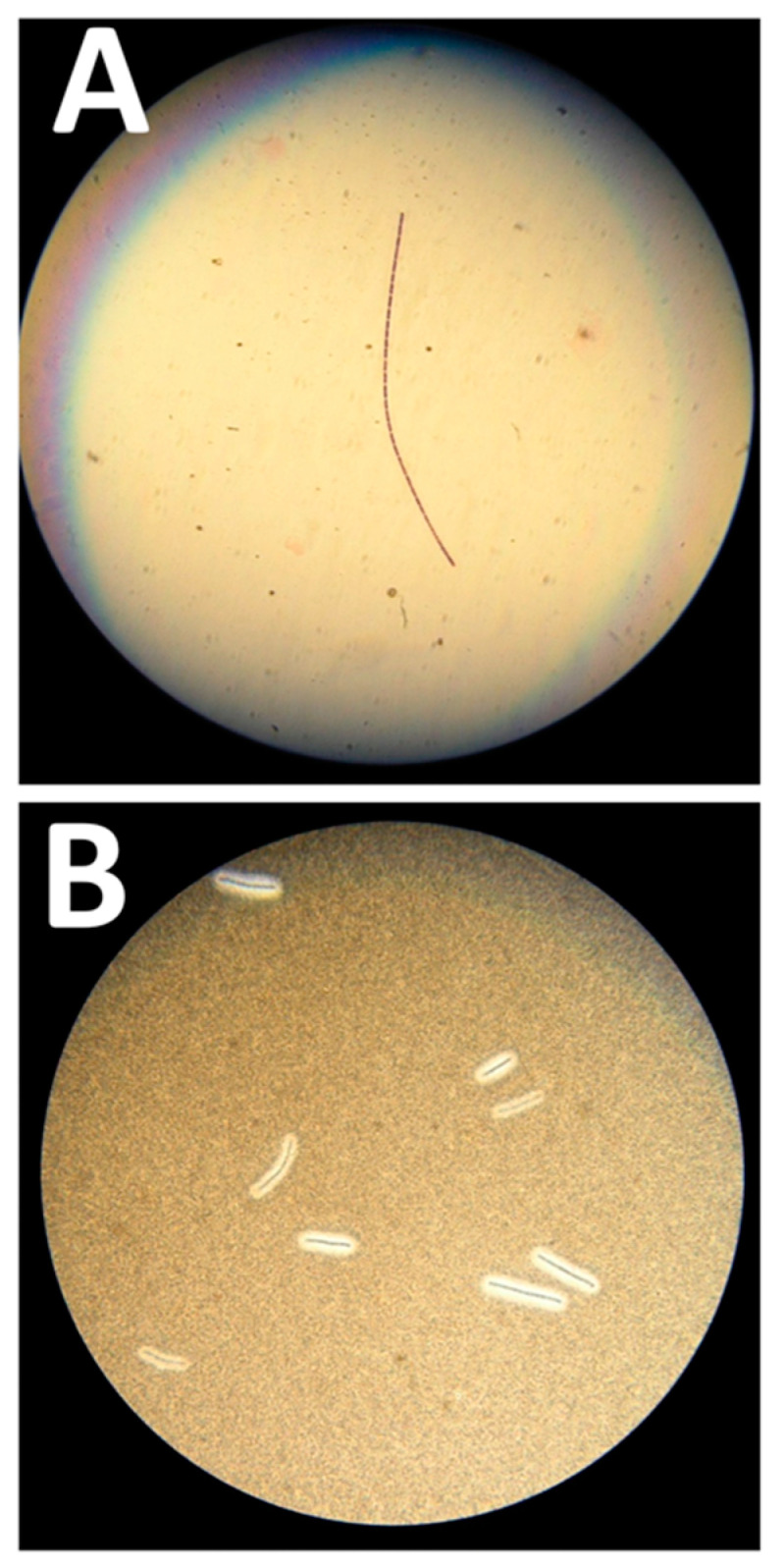
Microbiological features of the *Bacillus anthracis* strain. (**A**) Gram-positive “jointed bamboo-rod” bacilli forming long chains pathognomonic of *B. anthracis*. (**B**) Black ink negative staining of the *B. anthracis* bacilli capsule. The capsule repels the ink around the bacilli chains.

**Figure 2 microorganisms-08-00985-f002:**
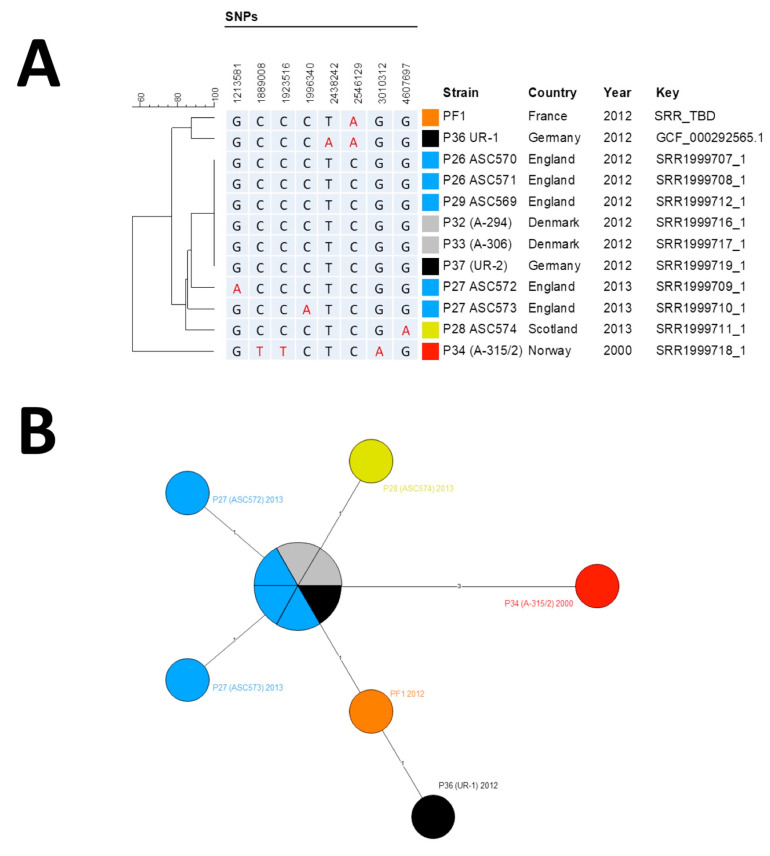
Phylogenetic analysis of *B. anthracis* strains. (**A**) Phylogenic tree of single-nucleotide polymorphism (SNP) profiles found among Group II strains. Position is calculated from Ames Ancestor Reference Genome. (**B**) Minimum spanning tree based on SNP profiles among Group II strains. (**A**,**B**) The different strains groups are color-coded by their geographical origin: red, original Norwegian strain; orange, French strain described here; light blue, English strains; yellow, Scottish strain; black, German strains; gray; Danish strains.

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
