# Peer review of "Case Report of an Injectional Anthrax in France, 2012"

_microorganisms, 2020, doi:10.3390/microorganisms8070985_

Round 1

Reviewer 1 Report

The authors Thouret et al. submitted a short case report of injectional anthrax in France. The manuscript shows a good description about the clinical case including clinical parameters and combines this with the diagnostic laboratory results. For diagnostic the methods are appropriate and state-of-the-art. But as obviously, based on the data, it is an cold case from a point of actuality, which is now investigated by NGS and phylogenetic analyses. Nevertheless the provided sequence bridge the gap between the isolates of the european outbreak in 2012. And that is, why this study is worthy for publication.

Some minor issues:

Titel: please include the year 2012.

Abstract: please delete "Interestingly" in the last sentence. If the sample is from 2012 in France, it is not surprising or interesting, that it belongs into this group. it is  - as supposed.

Results 3.1: a) When did the final identification took place? b) Which "in-house" real-time PCR? Be more specific. Was this a validated PCR? Is this PCR really specific? At least, specifiy the genomic targets of the PCR. In the case, that both of the information are missing, let somebody suppose, that something was unspoken. Maybe, the final identification took place several weeks after the patient was discharged from the ICU? or years after?

Discussion: The section starts with "Eventually" and ends with "Eventually". Please rephrase both. For the last sentence, just delete it, as the statement is just right.

Author Response

We appreciate the reviewer comments. As suggested by the reviewer, we have substantially modified our manuscript and provide below an answer to all comments.

Titel: please include the year 2012.

Title has been modified accordingly

Abstract: please delete "Interestingly" in the last sentence. If the sample is from 2012 in France, it is not surprising or interesting, that it belongs into this group. it is  - as supposed.

The abstract has been modified.

Results 3.1: a) When did the final identification took place? b) Which "in-house" real-time PCR? Be more specific. Was this a validated PCR? Is this PCR really specific? At least, specifiy the genomic targets of the PCR.

The final identification took place shortly after getting out of ICU. We have indicated this in the text.

3 genes are targeted by our in-house PCR: pagA, capC and SapA. The presence of the 3 genes were targeted, conjunction of both targets indicating the presence of a fully virulent B. anthracis strain. For confidentiality reason we can not provide the primer sequences.

Discussion: The section starts with "Eventually" and ends with "Eventually". Please rephrase both. For the last sentence, just delete it, as the statement is just right.

The word "eventually" has been deleted.

Reviewer 2 Report

It is interesting, although quite late (case was in 2012), to get some information about the only injectional anthrax case in France and about the corresponding strain. This untypical and very rare form of anthrax might appear again and it would be good for clinicians to keep it in mind.

I would like to get some more information about the microbiological background. The strain was initially identified as B. cereus which is normally haemolytic on blood agar, whereas B. anthracis is non-haemolytic. Was this important difference observed in this case? If the strain was identified biochemically as B. cereus, but non-haemolytic, further tests should have been performed immediately.

The 16S rRNA sequences are very similar, if not identical, in the B. cereus group. Why did 16S rRNA sequencing point to B. anthracis? A reference should be indicated where the differences are explained.

Author Response

We appreciate the reviewer comments. As suggested by the reviewer, we have substantially modified our manuscript and provide below an answer to all comments.

I would like to get some more information about the microbiological background. The strain was initially identified as B. cereus which is normally haemolytic on blood agar, whereas B. anthracis is non-haemolytic. Was this important difference observed in this case? If the strain was identified biochemically as B. cereus, but non-haemolytic, further tests should have been performed immediately. 

When we got the strain at the National reference lab, at the examination, we observed Gram-positive “jointed bamboo-rod” bacilli. The bacilli were non-hemolytic, immobile and capsulated, which suggested strongly B. anthracis. We did not get information on what was done before. 

We have modified the text accordingly

The 16S rRNA sequences are very similar, if not identical, in the B. cereus group. Why did 16S rRNA sequencing point to B. anthracis?

This is a good point. 16S rRNA PCR sequencing confirmed the presence of a Bacillus cereus group member (including B. anthracis). The strain has been addressed to us for as a suspicion. We have modified the text accordingly.